# Delta-Procalcitonin and Vitamin D Can Predict Mortality of Internal Medicine Patients with Microbiological Identified Sepsis

**DOI:** 10.3390/medicina57040331

**Published:** 2021-04-01

**Authors:** Alberto Tosoni, Anthony Cossari, Mattia Paratore, Michele Impagnatiello, Giovanna Passaro, Carla Vincenza Vallone, Vincenzo Zaccone, Antonio Gasbarrini, Giovanni Addolorato, Salvatore De Cosmo, Antonio Mirijello

**Affiliations:** 1CEMAD Digestive Disease Center, Fondazione Policlinico Universitario “A. Gemelli” IRCCS, Università Cattolica del Sacro Cuore, 00168 Rome, Italy; mattia_paratore@virgilio.it (M.P.); mikimpa@libero.it (M.I.); antonio.gasbarrini@unicatt.it (A.G.); giovanni.addolorato@unicatt.it (G.A.); 2Department of Economics, Statistics and Finance “Giovanni Anania”, University of Calabria, 87036 Rende, Italy; anthony.cossari@unical.it; 3Department of Geriatrics, Fondazione Policlinico Universitario “A. Gemelli” IRCCS, 00168 Rome, Italy; passaro.giovanna@gmail.com; 4Department of Emergency and Critical Care, Azienda Ospedaliera Universitaria San Giovanni di Dio e Ruggi D’Aragona, 84125 Salerno, Italy; carlavallone@hotmail.it; 5Department of Internal and Subintensive Medicine, Azienda Ospedaliero-Universitaria “Ospedali Riuniti”, 60126 Ancona, Italy; vincenzozaccone@libero.it; 6Department of Medical Sciences, IRCCS Casa Sollievo della Sofferenza, 71013 San Giovanni Rotondo, Italy; s.decosmo@operapadrepio.it

**Keywords:** procalcitonin kinetics, prognostication, sepsis biomarkers

## Abstract

*Background*: The management of septic patients hospitalized in Internal Medicine wards represents a challenge due to their complexity and heterogeneity, and a high mortality rate. Among the available prognostic tools, procalcitonin (PCT) is considered a marker of bacterial infection. Furthermore, an association between vitamin D deficiency and poor sepsis-related outcomes has been described. *Objectives:* To evaluate the prognostic accuracy of two consecutive PCT determinations (Delta-PCT) and of vitamin D levels in predicting mortality in a population of patients with microbiological identified sepsis admitted to Internal Medicine wards. *Methods*: This is a sub-analysis of a previous prospective study. A total of 80 patients had at least two available consecutive PCT determinations, while 63 had also vitamin D. Delta-PCT was defined as a reduction of PCT > 50% after 48 h, >75% after 72 h, and >85% after 96 h. Mortality rate at 28- and 90-days were considered as main outcome. *Results:* Mortality rate was 18.7% at 28-days and 30.0% at 90-days. Baseline PCT levels did not differ between survived and deceased patients (28-days: *p =* 0.525; 90-days: *p =* 0.088). A significantly higher proportion of survived patients showed Delta-PCT (28-days: *p =* 0.002; 90-days: *p* < 0.001). Delta-PCT was associated with a lower 28-days (*p =* 0.007; OR = 0.12, 95%CI 0.02–0.46) and 90-days mortality (*p =* 0.001; OR = 0.17, 95%CI 0.06–0.48). A significantly higher proportion of deceased patients showed severe vitamin D deficiency (28-days: *p =* 0.047; 90-days: *p =* 0.049). Severe vitamin D deficiency was associated with a higher 28-days (*p =* 0.058; OR = 3.95, 95%CI 1.04–19.43) and 90-days mortality (*p =* 0.054; OR = 2.94, 95%CI 1.00–9.23). *Conclusions*: Delta-PCT and vitamin D represent two useful tests for predicting prognosis of septic patients admitted to Internal Medicine wards.

## 1. Introduction

Sepsis is a leading cause of death and health care systems major burden worldwide [1]. Sepsis has been estimated to cause about half of all deaths occurring in hospitals [2]. The prevention, diagnosis, and treatment of sepsis should be considered a global health priority according to the World Health Assembly. 

Medical tools available for the management of septic patients, and used in daily clinical practice, have been mainly developed in Intensive Care Units (ICUs) and have not been extensively validated in Internal Medicine (IM) wards [3]. Among the evaluated scores, only a few showed good reliability in predicting mortality [4]. In particular, the accuracy of these scores could be low or inadequate for IM patients; this population is characterized by high heterogeneity, advanced age, and multiple comorbidities impacting on prognosis [5]. At present, there is uncertainty about the optimal clinical score to be used for septic IM patients.

Procalcitonin (PCT) is a polypeptide released after the interaction between cytokine-activated macrophages and endothelial cells in response to bacterial components, particularly lipopolysaccharide [6]. Circulating PCT levels rapidly raise during the early phase of sepsis, reaching a peak value proportionally correlated with the severity of bacterial infection and rapidly decrease, due to its short half-life, after the resolution of disease [7]. Thanks to these characteristics, PCT can be considered a fundamental marker for the recognition of bacterial infection and sepsis. Moreover, PCT could play a role as prognostic marker for predicting outcomes [8], and it can be used as a guide to antibiotic therapy, although not as a stand-alone test [9]. In fact, it must be considered that there are differences in the observed levels of PCT in relation to several variables (e.g., type and site of infection, host’s comorbidities, etc.) [10]. Most of the evidence in the literature suggests that baseline PCT levels are helpful in identifying the sickest patients, but not in predicting outcome. Multiple PCT determinations (PCT kinetics) appear to be more adequate for this purpose [11].

Vitamin D is a hormone playing its primary role in the bone homeostasis, but it is also involved in regulating immunity, both innate and adaptive [12]. The prevalence of vitamin D deficiency is particularly high among institutionalized subjects or those with other concomitant diseases [13]. As described in a previous study, the prevalence of vitamin D insufficiency was high in patients with bloodstream infection and sepsis admitted to Internal Medicine wards [4]. Low vitamin D levels can also be associated with a worse prognosis in patients with sepsis, but the results of studies performed in ICUs are heterogeneous [14,15,16].

The aim of the present study was to evaluate the accuracy of two consecutive PCT determinations (Delta-PCT) and vitamin D levels in predicting mortality, among a population of IM inpatients with microbiological identified sepsis.

## 2. Patients and Methods

### 2.1. Patients

The Internal Medicine Sepsis Study Group has promoted a 12-months sepsis surveillance program in two Internal Medicine Units of the “Agostino Gemelli” University Hospital, Catholic University of Rome, Rome, Italy [4]. Sepsis was defined according Sepsis-2 definition [17], while Quick SOFA (qSOFA) score was calculated according to Sepsis-3 definition [18]. During a screening phase, clinical information, laboratory data (including PCT and vitamin D), and clinical scores of 226 consecutive patients were recorded. Successively, after excluding patients with negative blood cultures, absence of SIRS criteria or non-clinically significant pathogens isolated on blood cultures, a total of 88 microbiologically-identified septic patients were included in the main study [4]. A total of 80 patients had at least two consecutive PCT determinations, thus they represent the sample evaluated in the present paper for statistical purposes.

### 2.2. Methods

This is a sub-analysis of a database including prospectively collected data of a cohort of consecutive patients with microbiological-identified sepsis admitted to an IM Unit [4]. The study was conducted according to local Ethical Committee guidelines. Anonymized clinical data were extracted from clinical records and recorded; thus informed consent was waived due to the observational, non-interventional design of the study. 

Data regarding 28- and 90-days mortality were retained from the initial study [4].

PCT was assessed at the time of collection of blood cultures (T0, baseline) and during antimicrobial treatment (T1, 48–96 h from baseline). As previously reported, a PCT level > 2 ng/mL was considered a significant cut-off for sepsis. Delta-PCT was calculated as the percentual variation of PCT at T1 compared to T0. We defined Delta-PCT as a reduction of PCT levels > 50% after 48 h or >75% after 72 h or >85% after 96 h from T0. Thus, a reduction of PCT levels lower than these cut-offs or a baseline PCT < 2 ng/mL was considered as “absence of Delta-PCT.”

Vitamin D assay was available in 63 out 80 (78.7%) patients. Although the Endocrine Society suggests specific categories for different vitamin D levels (e.g., vitamin D deficiency: 25(OH)D ≤ 19.9 ng/mL; vitamin D insufficiency: 20–29.9 ng/mL; vitamin D normal group: ≥30 ng/mL) [19], we adopted a dichotomous classification based on the presence of severe vitamin D deficiency (<7 ng/mL). This choice was done on the basis of literature data showing an association between severe vitamin D deficiency and mortality, in both critically ill and non-critically ill patients [15,20,21]. 

### 2.3. Statistical Analysis

A number of statistical procedures were applied for analysis of data, both descriptive and inferential, including numerical summaries (reported throughout the paper), Wilcoxon tests, Chi-square tests of independence, and logistic regression. The ultimate goal of the analysis was to primarily study the effect of Delta-PCT on 28-days and, respectively, 90-days mortality in a logistic regression framework. Moreover, vitamin D deficiency was studied in a similar way. First, a two-sample Wilcoxon test was run for assessing location differences of PCT levels at T0 between the group of deceased versus that of non-deceased patients. Then, a standard chi-squared test of independence in a two-way contingency table was used for tentatively testing the influence of both Delta-PCT and vitamin D deficiency on mortality. Afterwards, logistic regression was performed for an in-depth study of the effect of Delta-PCT as well as vitamin D deficiency on mortality. Obviously, all these analyses were repeated for both 28- and 90-days mortality. For the correct application of logistic regression, standard model checking techniques were run to assess model adequacy and thus validate the analysis method. All the computations were carried out by using the free software R [22]. 

## 3. Results

Main characteristics of the studied population have already been described [4] and are summarized in Table 1. Median age of patients was 75 years old. A significant proportion of them had a history of immunosuppression (40.9%), neoplasm (39.3%), diabetes (35.9%), or end-stage illness (23.9%). The majority of patients had received an antibiotic treatment in the previous 6 months (70.4%). 

A total of 15 out 80 patients (18.7%) died at 28-days and 27 out 80 (30.0%) died at 90-days. Baseline PCT levels did not differ between survived and deceased patients. A total of 39 patients (48.7%) showed Delta-PCT. A total of 31 out 63 patients (49.2%) patients showed severe vitamin D deficiency. Vitamin D levels were significantly lower in deceased than survived patients.

### 3.1. PCT and Mortality

The two-sample Wilcoxon tests showed that location values (specifically medians) of PCT levels at T0 did not differ between survived and deceased patients at 28-days (4.03 vs. 2.26; *p* = 0.525) nor at 90-days (5.54 vs. 1.83; *p* = 0.088). The Chi-square tests of independence showed a significantly higher proportion of patients with Delta-PCT among survivors at 28-days (*p* = 0.002) (Table 2) and at 90-days (*p* < 0.001) (Table 2). The accuracy of Delta-PCT (sensitivity, specificity, positive, and negative predictive values) in predicting 28- and 90-days patient’s mortality are reported in Table 2. The logistic regression showed that the presence of Delta-PCT was associated with a lower 28-days (*p* = 0.007; OR = 0.12, 95%CI 0.02–0.46) and 90-days mortality (*p* = 0.001; OR = 0.17, 95%CI 0.06–0.48).

### 3.2. Vitamin D and Mortality

The Chi-squared tests of independence showed a higher proportion of patients with severe vitamin D deficiency among deceased patients at 28-days (*p* = 0.047) (Table 3) and at 90-days (*p* = 0.049) (Table 3). The accuracy of vitamin D (sensitivity, specificity, positive, and negative predictive values) in predicting 28- and 90-days patient’s mortality is reported in Table 3. The logistic regression showed that severe vitamin D deficiency was associated with a higher 28-days (*p* = 0.058; OR = 3.95, 95%CI 1.04–19.43) and 90-days mortality (*p* = 0.054; OR = 2.94, 95%CI 1.00–9.23), even if the level of significance was borderline.

### 3.3. Two-Factors Logistic Regression Analysis

Finally, a two factors logistic regression confirmed previous results in terms of a negative effect of Delta-PCT on mortality both at 28-days (*p* = 0.012, OR = 0.06, 95%CI 0.003–0.38) and 90-days (*p* = 0.003, OR = 0.13, 95%CI 0.03–0.46) and a positive effect of severe vitamin D deficiency on mortality both at 28-days (*p* = 0.056; OR = 4.43, 95%CI 1.05–23.82) and 90-days (*p* = 0.053; OR = 3.31, 95%CI 1.02–11.83).

## 4. Discussion

The present study shows that Delta-PCT represents an independent predictor of outcome in a cohort of IM patients affected by bloodstream infection and sepsis. Literature data on prognostic performances of PCT in the IM setting are few and controversial. 

We previously showed in the main cohort of this study that baseline PCT did not predict mortality [4]. In line with our results, Papadimitriou-Olivgeris and colleagues showed that baseline PCT levels did not differ between survivors and non-survivors and PCT was not an independent predictor of mortality in a cohort of patients with similar characteristics [23]. However, although similar, IM patients are likely to show a high heterogeneity limiting the generalizability of results derived by single cohorts [5]. 

The prognostic performances of Delta-PCT (PCT kinetics) in the IM setting have been evaluated by a few studies. To the best of our knowledge, the study by Pieralli and colleagues is the only one conducted in a non-ICU setting, aiming to evaluate the role of PCT kinetics in predicting 30-days mortality in a sample of 144 patients with severe sepsis and/or septic shock admitted to Emergency Departments (EDs) or general wards [24]. As main result, Delta-PCT independently predicted 30-day mortality. The present study confirms that repeated PCT determinations with Delta-PCT assessment could be a useful tool to assess both 28- and 90-days risk of mortality in IM septic patients. 

The ability of Delta-PCT to be a better prognostic marker than single PCT measurement has several explanations. First, the value of a single PCT measurement as a predictor of outcome is poor given the large overlap between false negative and false positive values, different normal ranges and high interindividual variability due to acute comorbid states [5,25]. Moreover, in patients with BSIs, PCT levels depend on the etiological agent, being significantly higher in Gram-negative BSI than in Gram-positive or Candida BSIs, although PCT is insufficient to make an etiologic diagnosis when used alone [10]. Persistently high PCT values may indicate a persistence of the infectious state and/or a reduced response to antibiotic treatment. In addition, it should be considered that patients with an infection and impaired responsiveness of the immune system could show persistently low PCT values [26]. In both cases, the absence of a Delta-PCT could indicate an early risk of mortality. On the contrary, Delta-PCT often correlates with clinical improvement of the infectious picture. It can be used as a marker of efficacy of antibiotic treatment, even for an early de-escalation in order to reduce antibiotics side effects, as demonstrated in ICU studies [27,28]. 

In the present sample of IM septic patients, the accuracy of Delta-PCT was better in terms of specificity/PPV, than sensitivity/NPV. In other words, it is a biomarker with better “rule-out” than “rule-in” performances. 

The present study confirms our previous observation on severe vitamin D deficiency as independent predictor of death. This observation is in line with a recent meta-analysis, that showed that lower vitamin D at admission was independently associated with increased risk or mortality in patients with sepsis, even applying different diagnostic criteria for sepsis (SIRS, Sepsis-2, or Sepsis-3) [29]. However, this observation requires future epidemiological studies to understand whether low vitamin D levels represent a causal factor for sepsis due to reduced immune function or an epiphenomenon due to increased tissue utilization associated with inflammation [25]. Vitamin D is able to induce the expression of antibacterial proteins and to enhance the environment in which they function [30]. Thus, the increased susceptibility to infections among patients with vitamin D deficiency could be explained by reduced bacterial killing activity in several cell types. Severe vitamin D deficiency could also be directly involved in “freezing” the individual’s immune response capacity, in the context of sepsis itself [26]. This observation could confirm the susceptibility of Internal Medicine patients (e.g., comorbid, elderly, and institutionalized) with severe vitamin D deficiency to infections, sepsis, and poor sepsis-related outcomes. However, the observed accuracy of vitamin D deficiency was “low”.

Limitations of the present study are represented by the small sample size and the monocentric IM population. Moreover, results of our analysis have not been adjusted for potential confounders (e.g., age and sex), given their lack of influence observed in the main paper [4]. In any case, our observations need to be validated on a larger sample.

## 5. Conclusions

Sepsis is increasingly diagnosed in Internal Medicine wards. The management of septic patients with multiple comorbidities represents a real challenge due to the complexity of the syndrome and the high heterogeneity of septic populations. Even few and relatively easy assessments of biomarkers can be of help for patients’ outcome in certain conditions. Within this context, Delta-PCT and vitamin D could play a promising role for predicting the prognosis of septic patients admitted to Internal Medicine wards. 

## Figures and Tables

**Table 1 medicina-57-00331-t001:** Clinical characteristics of the 80 evaluated patients expressed as median (IQR; range); median values of survived or deceased patients at 28- and 90-days and statistical comparison.

	Study Patients (*n* = 80)	Survived 28-Days (*n* = 65)	Deceased 28-Days (*n* = 15)	*p*-Value	Survived 90-Days (*n* = 53)	Deceased 90-Days (*n* = 27)	*p*-Value
Male sex	46 (57.5%)	–	–	–	–	–	–
Age (years)	75 (64–83; 39–90)	72	83	0.068	73	81	0.032
BMI	24.7 (22.1–27.6; 19–44)	25	23	0.087	24.9	24.3	0.440
Procalcitonin (ng/mL)	2.91 (0.6–17.9; 0.1–100)	4.03	2.26	0.525	5.54	1.83	0.088
Vitamin D (ng/mL)	<7 (<7–9.9; <7–55.7)	7.7	<7	0.044	8.5	<7	0.032

**Table 2 medicina-57-00331-t002:** Number and percentage of patients survived or deceased at 28-days and at 90-days according to the presence of Delta-procalcitonin (DELTA-PCT) and statistical significance at chi-square tests of independence. (Sens = sensitivity; Spec = specificity; PPV = positive predictive value, NPV = negative predictive value).

**Table 2—28-days**	**Survived**	**Deceased**	**Total**	
Delta-PCT = 1, n (%)	37 (94.9)	2 (5.1)	39	PPV 0.95 (0.87–0.98)
Delta-PCT = 0, n (%)	28 (68.3)	13 (31.7)	41	NPV 0.32 (0.22–0.43)
Total	65	15	80	
	Sens. 0.57 (0.45–0.68)	Spec. 0.87 (0.62–0.96)		*p* = 0.002
**Table 2—90 days**	**Survived**	**Deceased**	**Total**	
Delta-PCT = 1, n (%)	33 (84.6)	6 (15.4)	39	PPV 0.85 (0.74–0.91)
Delta-PCT = 0, n (%)	20 (48.8)	21 (51.2)	41	NPV 0.51 (0.40–0.62)
Total	53	27	80	
	Sens. 0.62 (0.49–0.74)	Spec. 0.78 (0.59–0.89)		*p* < 0.0001

**Table 3 medicina-57-00331-t003:** Number and percentage of patients survived or deceased at 28-days and at 90-days according to the presence of vitamin D deficiency (>/<7 ng/mL) and statistical significance at chi-square tests of independence. (Sens = sensitivity; Spec = specificity; PPV = positive predictive value, NPV = negative predictive value).

**Table 3—28-days**	**Survived**	**Deceased**	**Total**	
Vitamin D > 7 ng/mL, n (%)	29 (90.6)	3 (9.4)	32	PPV 0.91 (0.80–0.96)
Vitamin D < 7 ng/mL, n (%)	22 (71.0)	9 (29.0)	31	NPV 0.29 (0.19–0.42)
Total	51	12	63	
	Sens. 0.57 (0.43–0.69)	Spec. 0.75 (0.47–0.91)		*p* = 0.046
**Table 3—90-days**	**Survived**	**Deceased**	**Total**	
Vitamin D > 7 ng/mL, n (%)	25 (78.1)	7 (21.9)	32	PPV 0.78 (0.66–0.87)
Vitamin D < 7 ng/mL, n (%)	17 (54.8)	14 (45.2)	31	NPV 0.45 (0.33–0.58)
Total	42	21	63	
	Sens. 0.60 (0.44–0.73)	Spec. 0.67 (0.45–0.83)		*p* = 0.049

## Data Availability

Data supporting reported results may be provided on reasonable request.

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
