# Peer review of "Delta-Procalcitonin and Vitamin D Can Predict Mortality of Internal Medicine Patients with Microbiological Identified Sepsis"

_medicina, 2021, doi:10.3390/medicina57040331_

Round 1

Reviewer 1 Report

The authors submitted an interesting paper reporting a secondary analysis of a single-centre prospective observational study, with the aim of evaluating the role of PCT and vitamin D levels in septic patients and their association with mortality.

The topic is interesting, but some issues need to be addressed by the authors.

I hope my comments would be of help to improve the quality of the manuscript.

Below, my specific comments by section.

INTRODUCTION

  • The authors should add a background on Vitamin D and sepsis to further clarify the rationale for the study.
  • The aim should also include Vitamin D as appropriate.

METHODS

  • Why did the authors decide to categorize Delta PCT in pre-specified groups? Why did they not consider it as a continuous variable for this analysis?
  • Why did the authors decide not to provide both the categorization for vitamin D (the one proposed by the society and the one coming from (more recent? More specific?) literature? Maybe both the analysis can provide interesting information
  • It is not clear to me whether the authors performed adjusted analysis for potential confounders. If so, please specify which covariates were included in which model. If not, this should be clearly listed among the limitations of the study.

RESULTS

  • Line 119. I suggest removing this phrase
  • I suggest presenting the baseline characteristics of the cohort of patients as Table 1.
  • Please do not use the term "influenced" or "influence" to present your results, as it may suggest a direct cause-effect relationship. I suggest considering your results as potential associations with risk.

DISCUSSION

  • Limitations of the study should be widened (please also see my previous comment in the methods section).
  • Recent literature on the potential role of PCT in differential microbiological diagnosis may provide another insight for discussion and comparisons (https://doi.org/10.1186/s13054-019-2481-y)
  • Please also widen the discussion on literature regarding the role of Vitamin D.

Author Response

We thank the Reviewer for his comments, which have improved both quality and readability of the manuscript. We have answered to all comments and marked correction in red along the text.

INTRODUCTION

  1. We added an adequate background on vitamin D and its controlling role on the immune system, underlining the possible correlation between low vitamin D values and worse prognosis in hospitalized patients with sepsis.
  2. We modified the presentation of the aim of the study, now including the vitamin D evaluation.

METHODS

  1. We thank the reviewer for this speculative observation. We decided to categorize Delta PCT in pre-specified groups in order to reflect data already published in the literature. Moreover, the concept of “delta-PCT” requires a categorization of PCT reduction.
  2. We opted to categorize vitamin D basing on the severe deficiency given that the majority of our patients (at least 50%) belonged to this category. Thus, a second categorization was not informative. Moreover, this cut-off was an independent predictor of mortality in the main paper.
  3. Co-variates (e.g. age and gender) were not included in the analysis given that they were not influencing the results in the main subsets. A specific sentence has been added among limitations, as suggested by the reviewer.

RESULTS

  1. We understand the Reviewer’s suggestion. However, we would prefer to maintain this sentence (line 119) in order to explain to readers the reason for this choice.
  2. Table 1 summarizing baseline characteristics of patients has been added.
  3. According to the suggestion, the term “influence” has been replaced with “association”.

DISCUSSION

  1. A specific sentence has been added among limitations, as suggested by the reviewer.
  2. As suggested by the Reviewer, we expanded the discussion with the possible role of PCT in the microbiological differential diagnosis, emphasizing how the single PCT value may also be influenced by the etiologic agent of septic condition.
  3. We expanded the discussion on vitamin D, adding references to recent literature and a reflection on the possible pathophysiological mechanisms underlying the correlation between vitamin D levels and poorer prognosis in Internal Medicine patients.

Reviewer 2 Report

The paper describes the data collection and analysis of 2 different biomarkers of sepsis, PCT and vitamin D as per SoC in an Internal Medicine ward. Both biomarkers have been extensively studied and validated in sepsis and in this respect the study as presented doesn't appear to provide an added value in the sepsis biomarkers field of research.

Author Response

Dear Editor,

We’ve read the comments provided by the Reviewer 2: in his opinion, our paper does not add any value to the sepsis field, since – in his opinion – PCT and Vitamin D represent the standard of care in Internal Medicine wards. We fully respect these thoughts. Basing on his comments, we understand that the Reviewer 2 is working in a highly specialized research Institute, where every innovation and speculation coming from the literature is immediately transferred to the routine daily clinical practice. For these reasons we congratulate with him.

On the contrary, according to our knowledge and to our real-life practice of several secondary and tertiary care hospitals, the use biomarkers for sepsis in Internal Medicine is often contrasting. In particular, repeated assays of PCT are performed, but the real value of DELTA-PCT is underestimated or ignored. This is the reason that led us to perform a sub-analysis of our previously published data showing the lack of utility of baseline PCT for prediction of mortality. Unsurprisingly, it’s true!, but in line with literature, the DELTA-PCT is a predictor of antibiotic response and mortality. This is a little information, confirming existing data, and making them more solid. As per the use of vitamin D for the prediction of mortality in Internal Medicine septic patients, I would suggest to take a survey among Internists or even Intensivists or Endocrinologists asking them about their use of vitamin D levels out of the calcium-phosporus-bone metabolism.

Finally, according to our opinion, the revision of a paper should improve its readability, utility in clinical practice and substance. We did not find any of these suggestions in the revision provided by the Reviewer 2.

We thank the Editors that gave us the possibility to revise our paper that, according to the reviewer 2, would have been rejected without any improvement.

Reviewer 3 Report

Tosoni and co-authors conducted a sub-analysis of a previous prospective study, investigating the prognostic accuracy of delta-PCT and vitamin D level in predicting mortality in a population of septic patients admitted to internal medicine wards. They found that delta-PCT and vitamin D are two useful markers for predicting prognosis in septic patients. Procalcitonin has been proposed as a useful tool to characterize systemic inflammation, infection, and sepsis. Moreover, findings from several randomized controlled trials indicate that the use of a PCT-guided antibiotic treatment algorithm is likely to reduce antibiotic exposure in septic patients, without an adverse effect on health outcomes. Alongside, severe vitamin D deficiency is an independent predictor of death in critically ill patients.

The manuscript is well written and data are nicely presented. No major relevant issues are present. However, there are a few phrasing and reference issues that should be revised by the authors in order to improve their work:

  1. Line 41: “a major burden on the health system” please, rewrite as “and health care systems major burden”
  2. Line 42: “Sepsis has been estimated to cause or contribute to about half of all deaths that occur in hospitals” please, rewrite as “Sepsis has been estimated to cause about half of all deaths occurring in hospitals”
  3. Line 55-63: Authors should implement description of PCT importance with references related to PCT antimicrobial guidance and differences in PCT observed levels according to the type of infection occurring in critically ill patients (Cortegiani et al. Critical Care (2019) 23:190 https://doi.org/10.1186/s13054-019-2481-y; Bassetti M, Russo A, Righi E, Dolso E, Merelli M, D'Aurizio F, et al. Role of procalcitonin in bacteremic patients and its potential use in predicting infection etiology. Expert Rev Anti-Infect Ther. 2019;17:99–105)
  4. Line 120: please, avoid the use of “briefly” at the beginning of results presentation
  5. Line 178-181: please, rephrase in order to avoid misinterpretations (…”the only study conducted…”).

Author Response

We thank the Reviewer for his comments, which have improved both quality and readability of the manuscript.

We have answered to all comments and marked correction in red along the text.

  1. Line 41: the text has been rewritten as suggested.
  2. Line 42: the text has been rewritten as suggested.
  3. Line 55-63: we have widened the introduction as suggested, highlighting the possibility of using PCT as a guide to antibiotic therapy, and the differences observed in PCT levels according to a number of variables; we also cited the suggested literature.
  4. Line 120: the term “Briefly” has been avoided.
  5. Line 178-181: the sentence has been rephrased as suggested, in order to avoid misinterpretations.

Round 2

Reviewer 1 Report

I have no further comments

Author Response

We thank the reviewer for his revision.

Reviewer 2 Report

I agree with the Authors that the reported data  add little information, confirming existing data, and making them more solid. Anyway, of value for clinical practice.

The above does not change the relatively low scientific soundness of the paper. However, in the discussion it could be emphasized the concept that even few and relatively easy assessments of biomarkers can be of help for the patient's outcome in certain condition. With this revision in the discussion of the results i would be willing to approve the paper as otherwise presented. 

Author Response

We thank the reviewer for his suggestion. A specific sentence was added in the conclusion section and it has been marked in red.